# The Prediction of Calorific Value of Carbonized Solid Fuel Produced from Refuse-Derived Fuel in the Low-Temperature Pyrolysis in CO_2_

**DOI:** 10.3390/ma14010049

**Published:** 2020-12-24

**Authors:** Ewa Syguła, Kacper Świechowski, Paweł Stępień, Jacek A. Koziel, Andrzej Białowiec

**Affiliations:** 1Faculty of Life Sciences and Technology, Institute of Agricultural Engineering, Wrocław University of Environmental and Life Sciences, 37/41 Chełmońskiego Str., 51-630 Wrocław, Poland; ewa.sygula@upwr.edu.pl (E.S.); pawel.stepien@upwr.edu.pl (P.S.); andrzej.bialowiec@upwr.edu.pl (A.B.); 2Department of Agricultural and Biosystems Engineering, Iowa State University, Ames, IA 50011, USA; koziel@iastate.edu

**Keywords:** higher heating value, waste to energy, waste to carbon, municipal solid waste, waste conversion, waste recycling, thermal treatment, waste management, resource recovery

## Abstract

The decrease in the calorific value of refuse-derived fuel (RDF) is an unintended outcome of the progress made toward more sustainable waste management. Plastics and paper separation and recycling leads to the overall decrease in waste’s calorific value, further limiting its applicability for thermal treatment. Pyrolysis has been proposed to densify energy in RDF and generate carbonized solid fuel (CSF). The challenge is that the feedstock composition of RDF is variable and site-specific. Therefore, the optimal pyrolysis conditions have to be established every time, depending on feedstock composition. In this research, we developed a model to predict the higher heating value (HHV) of the RDF composed of eight morphological refuse groups after low-temperature pyrolysis in CO_2_ (300–500 °C and 60 min) into CSF. The model considers cardboard, fabric, kitchen waste, paper, plastic, rubber, PAP/AL/PE (paper/aluminum/polyethylene) composite packaging pack, and wood, pyrolysis temperature, and residence time. The determination coefficients (R^2^) and Akaike information criteria were used for selecting the best model among four mathematical functions: (I) linear, (II) second-order polynomial, (III) factorial regression, and (IV) quadratic regression. For each RDF waste component, among these four models, the one best fitted to the experimental data was chosen; then, these models were integrated into the general model that predicts the HHV of CSF from the blends of RDF. The general model was validated experimentally by the application to the RDF blends. The validation revealed that the model explains 70–75% CSF HHV data variability. The results show that the optimal pyrolysis conditions depend on the most abundant waste in the waste mixture. High-quality CSF can be obtained from wastes such as paper, carton, plastic, and rubber when processed at relatively low temperatures (300 °C), whereas wastes such as fabrics and wood require higher temperatures (500 °C). The developed model showed that it is possible to achieve the CSF with the highest HHV value by optimizing the pyrolysis of RDF with the process temperature, residence time, and feedstock blends pretreatment.

## 1. Introduction

### 1.1. Background of Current Situation

The market for alternative fuels such as RDF/SRF (refuse-derived fuel/solid recovered fuel) is developing dynamically, and therefore optimization processes of waste used in the energy industry are sought to increase the efficiency of the process. The RDF is an alternative fuel produced from the combustible fraction of municipal solid waste. Currently, in Poland, RDF is co-fired with other conventional fuels in cement kilns, and the annual consumption of alternative fuel is approximately 1.6 mln Mg [1,2]. The calorific value of RDF fuel is one of the key parameters determining the fuel quality and varies in the range of 15–21 MJ·kg^−1^. The next important criteria for the quality of RDF are ash content of ≈3.4–16% and moisture content, which must not exceed 20% [3].

The problem related to the use of alternative fuels derived from waste is its heterogeneity and variability. The composition of waste used as fuel in incineration or cement plants depends on many factors, such as the season of the year, waste collection and pretreatment, the wealth of the citizens, locations, and others. Due to the progressive implementation of zero waste policy in the EU, the residual waste becomes less valuable in the energy context (more high calorific plastic fraction is collected, reused, or recycled) [4]. Therefore, a residual waste upgrade method, before the incineration, becomes necessary to provide a high calorific fuel for plants.

Methods of conversion/upgrade of residual waste include mainly thermal conversions such as torrefaction, pyrolysis, gasification, hydrothermal liquefaction, and its hybrid modifications. The conversion of municipal solid waste into noble fuels by thermal conversion, i.e., gases, heating oils, gasoline, diesel, and CSF (carbonized solid fuel), is researched and already under consideration across Europe because the utilization of residual waste fits well to the circular economy concept [5,6,7]. The results from Slovakia show that the conversion of tires and plastics to fuel products is feasible [7]. In the work of Zabaniotou and Stavropoulos [8], the utilization of char from tires to fuel and valuable-added products (activated carbon) was investigated, and both pathways turned out to be feasible [8]. In recent years, the pyrolysis of a mixture of biomass and municipal solid waste in the CO_2_ atmosphere received attention [9,10,11,12]. The CO_2_-assisted pyrolysis increases the generation of useful gases (e.g., carbon monoxide), decreases the generation of tars, and increases the thermal cracking of volatile compounds, which leads to a decreased generation of harmful chemical compounds and makes pyrolysis more environmentally friendly [12].

### 1.2. Municipal Solid Waste Characterization

In general, the MSW is a mixture of highly diversified waste. Typical RDF made from MSW consists of a carton, fabric, kitchen waste, paper, plastic, rubber, PAP/AL/PE (paper/aluminum/polyethylene) composite packaging pack (multi-material packaging), and wood [13]. Thus, applying pyrolysis to MSW requires the optimization of process parameters and input substrate, depending on the process purpose.

Waste from cellulose components, such as cardboard and paper, represents ≈350,000 Mg in the waste stream collected selectively in Poland. The world produces ≈400 million Mg of paper and cardboard waste [14]. On a global scale, the paper recovery rate is ≈58% (≈70% in the EU). The waste paper can be recycled up to 4 times without a significant impact on the strength of the fibers. Nevertheless, that process will end, and finally, paper/cardboard has to be disposed of [15,16].

Textile waste accounts for 2% of the total waste stream in Poland (384,000 Mg), where one of the main components is cotton [17]. Currently, companies are being established to create alternative fuels from cotton waste because of the advantages of this material, such as low humidity, high calorific value, and low chlorine content [18,19].

About 1,200,000 Mg of biodegradable waste was produced in Poland in 2019. Kitchen waste is the main component of biodegradable waste, which accounts for approximately 36% of the total MSW stream [20]. The kitchen waste has ≈40% moisture, a low calorific value estimated at ≈10 MJ·kg^−1^, and a tendency to rooting and odor. This problem can be solved by pyrolysis, which leads to water removal and an increase in energy potential [21,22,23].

The plastics waste, which is one of the predominant components of RDF/SRF mixture (constitute ≈30–50%, sometimes reaching over 80%) [24] is made from other polymers than the above-mentioned RDF components, and its thermal degradation takes place at 400–600 °C (PE, PET, PE plastic). PVC material is less thermally stable and starts to be degraded at ≈250 °C [25]. In 2019, the selectively collected plastic waste in Poland amounted to ≈400,000 Mg [14]. Polyethylene accounts for a large share of the entire waste stream, which is due to the low price of the raw material and its wide use. Pyrolysis of polyethylene leads to an improvement in the parameters of the alternative fuel [26].

PAP/AL/PE composite packaging pack (tetra pack) has a diverse composition that is difficult to manage as a waste. PAP/AL/PE waste constitutes 1% of the waste stream in Poland. The main materials used to make tetra packs are paper (75% by weight), polyethylene (20%), and aluminum (5%) [27]. A common method of processing PAP/AL/PE waste is recycling, which produces paper products from cellulose-containing materials. PAP/AL/PE can also be used for making waterproof boards. In New Zealand, recycled PAP/AL/PE boxes are mixed with carbon fines and then formed into a briquette called “Hot Rocks” [28]. PAP/AL/PE waste also enables making alternative fuels and the recovery of aluminum [29,30].

Rubber waste accounts for 180,000 Mg in Poland annually. Rubber is not biodegradable, which makes it impossible to manage by biological methods. The treatment of rubber waste can be carried out mechanically or chemically [31]. The pyrolysis of rubber allows producing highly energetic solid fuel. Carbonized rubber’s calorific value is ≈31 MJ·kg^−1^; the moisture content reaches 1%, while ash constitutes ≈12% [32,33,34].

Wood waste in the MSW stream in Poland accounts for ≈53,000 Mg∙year^−1^. The range of wood waste management methods is quite wide, but the choice of appropriate technology is influenced by the degree of contamination [35]. Producing biochar from wood waste for the energetic purpose makes the most sense when the substrate is contaminated and other recycling methods are not suitable.

### 1.3. Proposed Solution—Low-Temperature Pyrolysis

In this work, low-temperature pyrolysis in CO_2_ was considered as a method for MSW components valorization by its carbonization. The main product of low-temperature pyrolysis is CSF. The CSF is characterized by lower humidity and higher energy density, resulting in a higher calorific value than raw waste. The energetic quality of CSF is mainly related to the quality of input materials and thermal process parameters such as temperature, residence time, or heating rate [36]. Low-temperature pyrolysis is a process of thermal breakdown of waste containing organic substances in the absence of oxygen and at 300–550 °C. The products of pyrolysis are pyrolytic gas, biochar, and a liquid fraction (a mixture of water and oils) [1]. The pyrolysis of waste is a very complex process in which many physical and chemical processes related to heat distribution and transport take place. The process of heat transfer from the source to the interior of the fuel takes place in various ways, including conduction, convection, or radiation [37]. Moreover, depending on the chemical reactions in the processed substrate, the heat transport direction can be inverted (exothermic reaction) [38]. The production of CSF also depends on the used substrate and its chemical composition. The different compositions result in various kinetics of thermal degradation, and therefore, each substrate has its optimal processing conditions to produce high-quality CSF [13]. The processing materials with different chemical properties may also result in overlapping reactions that can be positively or negatively synergetic for the process. Due to the above, there is a need to do experimental research to determine and predict the quality of CSF, especially when processed in a mixture of different types of waste components, as it is in the case of RDF.

### 1.4. Importance of Mathematical Approaches for Fuel Properties Predictions

The composition of RDF is highly variable; therefore, before the implementation of the RDF pyrolysis, it is difficult to execute the energy balance of the process and estimate the benefits from RDF conversion to CSF. It may require on-site specific experiments; however, analyzing the energetic properties of fuel made in the pyrolysis process by experiments is time-consuming and expensive. Therefore, the proposed solution is to use the empirical models based on elemental composition (C, H, N, S, O) such as Dulong- or Schuster-type equations [39] or less expensive proximate analysis (moisture content, ash, fixed carbon, and volatile matter) [40,41].

In previous work, we used a regression model based on torrefaction duration and temperature for fuel properties determination of torrefied biomass [42,43,44,45]. Other models that use process kinetics (mass loss) are available [46,47] to predict the properties of carbonized materials. In recent years, neural networks have also been used for the prediction of HHV [48].

Mathematical modeling of thermal processes is a useful engineering tool that can predict with reasonable accuracy the behavior of the tested material under various temperatures and process time without the need for experimental trials of every possible condition. The model based on experimental data can provide the optimal composition of RDF and process conditions under which the fuel has the best parameters, such as high heating value. The model for fuel properties determination is beneficial, especially for a mixture of fuel constituents representing many materials, precisely in the case of municipal waste streams [44,49,50,51].

Each of the above-mentioned methods allows for forecasting fuel parameters using experimental data. Nevertheless, each method requires a different methodology to obtain data that will be used to predict CSF quality. The main advantages and disadvantages of the most commonly used methods are summarized in Table 1. The most common and parsimonious method of higher heating value (HHV) predictions are equations based on the proximate analysis [40,41]. Then, more accurate predictions require elemental composition, needing more sophisticated equipment, and therefore higher cost [39]. The most precise prediction could be obtained via an artificial neural network approach. Artificial neural networks are used when there is a non-linear relationship between dependent and independent variables. Despite the increasing use of this forecasting method, it requires large databases, special software, and knowledge in programming languages. The main drawback of this method is limited possibilities for making useful conclusions about the impact of separate independent variables. The input data can result from the proximate and ultimate analysis and process parameters. These results can be considered together or separately [52,53].

In this work, the regression modeling was examined for four models based only on pyrolysis process parameters (temperature and residence time); proximate and ultimate data were not needed. The linear equation, second-order polynomial equation, factorial regression equation, and square regression equation were tested. The least-squares regression was used to estimate the parameters by minimizing the squared discrepancies between the experimental data and their expected values [56]. The purpose of forecasting using the mentioned models is to quantify the relationships between independent variables (temperature and time of the pyrolysis process) and the dependent variable (HHV). The linear equation (the simplest one) is a regression equation for a multiple regression model containing first-order effects for two continuous predictors (temperature and time). Polynomial regression models (here second-order) contain main effects and higher-order effects for continuous predictors but do not take into account interaction effects between the predictors. Factorial regression models are defined as ones in which all possible predictors multiplications are present [57]. In this study, the square regression equation is considered as a combination of second-order polynomial and factorial regression equations.

### 1.5. Research Aim

In this paper, for the first time, we provide regression models for HHV prediction for main morphological groups of RDF carbonized under low-temperature pyrolysis (300–500 °C, up to 60 min), and one general predictive model for the estimation of HHV of a CSF derived from RDF. The model needs readily available input data concerning the pyrolysis conditions (temperature and time) and the morphological fraction of main RDF components (by weight) to predict the HHV of CSF of blended waste feedstock. The mathematical regression method was chosen from the many methods available to predict the higher calorific value. The choice was supported by an adequate result in terms of modeling workload. Proposed inputs for mathematical modeling do not require complicated apparatus, and the needed analyses are carried out by weighing.

## 2. Materials and Methods

### 2.1. Experimental Procedure

The experimental procedure has been given in Figure 1. First, the individual components of RDF (cardboard, fabric, kitchen waste, paper, plastic, rubber, PAP/AL/PE, and wood) have been acquired from the market. These components were also used for RDF blends mixture preparation (RDF 1 and RDF 2). Each material was dried and then ground by knife mill through a screen of 1 mm. After that, materials that were not used immediately were stored in plastic containers in a freezer at −15 °C. Each RDF component and RDF blends were characterized for the following parameters: moisture content, organic matter content measured as a loss on ignition, ash, and combustible part content, elemental compositions (C, H, N, S, and O), and HHV. The RDF components and RDF blends were subjected individually to the pyrolysis process at different temperatures (300–500 °C) and residence time (20–60 min) at a CO_2_ gas flow of 2.5 dm^3^∙min^−1^. The CSF was produced under pyrolysis conditions concerning three independent variables: temperature, residence time, and type of the RDF component. The CSFs were analyzed for the same parameters as raw materials. The full experimental procedure and results have been given in our previous data descriptor article [58].

The results of the high heating value (HHV) of CSF (the dependent variable) made from each component of RDF were subjected to regressions analysis. The goal was to provide models that quantitatively describe the impact of temperature and residence time of pyrolysis on the high heating value of CSF produced from different components of RDF. There four regression equations were tested for each RDF component, among which the best fitted was chosen. The best equation/model for each component was selected based on the determination coefficient (R^2^) and the Akaike information criterion (AIC). Then, the best-fitted model for each RDF component was integrated into one overall (general) model that predicts the HHV of carbonized RDF. In the end, results calculated by the general model were validated by comparing with experimental data of two RDF mixtures pyrolyzed at the same conditions.

### 2.2. RDF Components and RDF Blends Preparation

The components of RDF were cardboard, fabric, kitchen waste, paper, plastic, rubber, PAP/AL/PE, and wood. The cardboard was represented by a gray carton, while fabrics were represented by a cotton t-shirt. The kitchen waste was a mixture of vegetables, 41.6%; banana peel, 29.7%; basic food, 22.29%; chicken meat, 0.2%; eggshells, 4%; and walnut shells, 2.2% (by weight). The paper was represented by office paper, plastic was represented by polyethylene foil (garbage bag), rubber was represented by an inner car tube, PAP/AL/PE was represented by Tetra Pak^®^-type packaging, and wood was represented by pruning tree branches. The RDF blends were made following the percentage share (by mass) presented in Figure 2. For more information about samples, please refer to the following data descriptor paper [58].

### 2.3. Low-Temperature Pyrolysis Process—Description

The low-temperature pyrolysis of RDF components and their blends was performed using a laboratory muffle furnace that simulated a fixed bed reactor. The process was conducted in the atmosphere of CO_2_. Then, 10 g of dry sample (components of RDF and its blends) were processed at temperatures of 300–500 °C and residence times at a setpoint temperature of 20–60 min. The carbonized RDFs (CSF) were removed and further analyzed after the reactor temperature decreased below 200 °C (to prevent the self-ignition of CSF). For detailed information about the CSF production procedure, please refer to [58].

### 2.4. HHV Model of Carbonized Solid Fuel—Determination

The results of the high heating value of carbonized solid fuels made from RDF components were subject to different regressions analyses (with different algebraic functions). Each RDF component was separately subjected to four model equations, and then the best-fitted one was chosen for a particular component (via R^2^ and AIC). The following regression models were considered: (I) multiple regression based on a linear model, (II) multiple regression based on the second-order polynomial model, (III) factorial regression, and (IV) quadratic regression (Table 2). Each model has intercept (a_1_) and regression coefficients (a_2_–a_6_). The analysis was performed using software StatSoft, Statistica 13.3 (TIBCO Software Inc., Palo Alto, CA, USA).

Determination coefficients (R^2^) for each model were used to estimate the degree of model matching to experimental data:(1)$$R^{2}=\overset{n}{\underset{i=1}{\sum }}{\left(\hat{y_{i}}-\bar{y}\right)}^{2}/\overset{n}{\underset{i=1}{\sum }}{\left(y_{i}-\bar{y}\right)}^{2}$$
where:
$R^{2}$—determination coefficients,i—subsequent observations,$\hat{y_{i}}$—value of the dependent variable predicted by the regression model,$\bar{y}$—mean value of the dependent variable (measured),$y_{i}$—value of the dependent variable (measured).

The AIC was used to select a better model among those with the same/similar R^2^. The AIC is used to select statistical models with a different number of predictors. In general, a model with more predictors gives more accurate predictions but is also more likely to experience over-fitting. The AIC assumes that the lower the value, the better the model (the model is simpler and has similar matching). The calculation of the AIC was performed by Equation (2).
(2)$$\mathrm{AIC}=n\cdot \ln(\sum_{i=1}^{n}e_{i}^{2})+2\cdot K$$
where:AIC—a value of Akaike analysis;*n*—the number of measurements;*e*—the value of the rest of the model;*K*—the number of regressions coefficients (including the intercept).

### 2.5. General Model of HHV of CSF from Mixtures RDF—Determination and Validation

The general model for the determination of HHV of CSF from RDF was created by combining the best fitting-models obtained according to the procedure described in Section 2.4 for particular RDF components. The general (overall) model can be written as:(3)$${\mathrm{HHV}}_{\mathrm{CSF}\;\mathrm{RDF}}\left(T,t\right)=\sum_{i}^{n}\left({\mathrm{HHV}}_{i}\left(T,t\right)\cdot \%share_{i}\right)/\sum_{i}^{n}\left(\%share_{i}\right)$$
where:
$HHV_{CSF\;RDF}\left(T,t\right)$—the estimated value of HHV of CSF from RDF at *T*&*t* conditions, MJ∙kg^−1^;$HHV_{i}\left(T,t\right)$—the estimated value of HHV of i-CSF from individual RDF component under *T*&*t* conditions, MJ∙kg^−1^;$\%share_{i}$—percentage share of i-CSF from individual RDF component in the total mass of CSF from RDF mixture, %.

The models were validated by using a linear correlation with the experimental data obtained from two carbonized RDF blends. The predicted values were compared with the experimental one using a root mean square error (*RMSE*). The *RMSE* is the standard deviation of the residuals:(4)$$RMSE={\left(\sum_{i=1}^{n}(z_{m}-z_{o})/n\right)}^{0.5}.\;$$
where:
RMSE—root mean square error;*n*—sample size/number of measurements;$z_{m}$—modeled value of HHV, MJ∙kg^−1^;$z_{o}$—observed, the experimental value of HHV, MJ∙kg^−1^.

## 3. Results and Discussion

The eight regression models for individual main components with the best fitting to experimental data were provided and described. These models can predict an HHV of the carbonized carton, fabric kitchen waste, paper, plastic, rubber, PAP/AL/PE composite packaging pack, and wood, based on pyrolysis temperature and residence time. Then, a general model for the prediction of HHV of carbonized RDF mixture was developed and validated with experimental data. The models’ regression coefficients for the particular waste component and general model for the mixture are available in the Appendix A.

### 3.1. CSF from Individual RDF Components Models

The results of the regression analysis are presented in Table 3. The four models were tested for each individual waste material. The models with the best fitting of the raw data are marked by bold font. Models I and III turned out to be less accurate and were not used. For carton, fabric, and paper, the better model was model II, whereas model IV was better for kitchen waste, plastic, rubber, PAP/AL/PE composite packaging pack, and wood. In the cases when the models had similar R^2^, the model with lower AIC was considered a better one.

The results show that the HHV of the carton did not change significantly with the increase of temperature and pyrolysis process time and was ≈16 MJ∙kg^−1^ (Figure 3a). The four selected models describing the HHV were characterized by a low R^2^ = 0.14. The lowest AIC value was observed in model II (AIC = 245.98) (Table 3).

For fabric, HHV increased with the temperature and time of the pyrolysis process in a relatively linear fashion (from ≈19 to 25 MJ∙kg^−1^) (Figure 3b). The selected model parameters fitting degree to experimental data did not exceed R^2^ = 0.53. However, the R^2^ values did not differ significantly (*p* > 0.05), from 0.51 to 0.53. Based on the AIC (416.63), model II was selected (Table 3) for further analysis.

In the case of the kitchen waste, the HHV of CSF slightly increased with the process temperature and time (from ≈19 to 21 MJ∙kg^−1^) (Figure 3c). Since model IV had the highest R^2^ = 0.64 (Table 3), the AIC value was not considered.

The results showed that the HHV of CSF produced from paper slightly decreased with the increase of temperature and time of the pyrolysis process. The CSF HHV decrease from ≈15 to <14 MJ∙kg^−1^ was observed (Figure 3d). The R^2^ varied from 0.29 to 0.36. The highest R^2^ values were obtained in models II and IV. Based on the AIC test, model II was selected (AIC = 250.05) (Table 3).

The thermally processed plastic decreased its HHV as the temperature and process time increased. In general, CSF HHV decreased from ≈35 to ~27 MJ∙kg^−1^ with temperature and time increase (Figure 3e). The highest R^2^ was in the case of model II and IV. Verification of the models using the AIC promoted model IV, with the lowest AIC = 483.70 (Table 3).

The HHV of CSF produced from the rubber decreased with an increase in process temperature from ≈35 to ≈12 MJ∙kg^−1^ (Figure 3f). The four proposed models did not differ significantly in R^2^ (0.84–0.87). The AIC of 404.39 indicated that model IV was the most suitable.

The HHV of CSF from the PAP/AL/PE composite packaging pack increased until ≈420 °C (from ≈21 to 30 MJ∙kg^−1^). At higher temperatures, a significant decrease of CSFs’ HHV was observed (Figure 3g). The R^2^ ranged from 0.01 to 0.72. Verification of the models with the AIC promoted model IV with AIC = 386.88 as the best one (Table 3).

Results for wood showed that its HHV increased with the temperature and time increase from ≈21 to 26 MJ∙kg^−1^ (Figure 3h), R^2^ = 0.76–0.78. The AIC promoted model IV (AIC = 254. 29) as the best one (Table 3).

The 3D presentation of the models (Figure 3) showed that for considered (individually) materials, the increase of process temperature and time of pyrolysis leads to different results of HHV of CSF. Lower process temperatures are more favorable to obtain the CSF with the highest possible HHV for carton, paper, plastic, and rubber. For these materials, the increase in temperature leads to decreasing HHV. The decrease in the case of plastic (polyethylene) material, and likely rubber as well, could be a result of pyrolysis performed in an open reactor (the pyrolysis gases were allowed to vent from the rector), and therefore, secondary reactions that favor biochar production did not take place. Tiikma et al. [59] reported that during the pyrolysis of polyethylene in a closed reactor, biochar yield decreased from 420 to 440 °C, while at >440 °C, it started to increase, whereas a liquid fraction showed the opposite trend. It means that the liquid starts to decompose to biochar and gas (secondary reactions) above 440 °C. It is worth noting that these reactions took place after ≈90 min and continued with longer residence time [59]. Knowing that the CSF produced for this study had a long cooling period from setpoint temperature to ambient (several hours), we assume that the reason for HHV decrease was due to the lack of secondary reactions.

The decrease of paper and carton (cellulose-based materials) HHV at higher temperatures is also interesting. Several works on the pyrolysis of cardboard and paper were published [60,61,62]. The results show that the HHV of CSF from cardboard/paper slightly increases with the temperature increase, but above a certain temperature, it starts to decline [60,61]. The reason for that could be due to the high ash content that increases with the process temperature [61]. The residence time also has a significant impact on HHV; in the above-mentioned works, materials were pyrolyzed at a residence time longer than 30 min. In the work of Sotoudehnia et al. [62], a short residence time was used (6.4 s), and the increase of pyrolysis temperature resulted in a slight increase of HHV for the CSF from cardboard. An additional reason for that phenomenon could be the loss of carbon due to the formation of acetic acid and its volatilization.

In the case of the PAP/AL/PE composite packaging pack (Figure 3g), the HHV behavior may be related to the lack of secondary reactions and to the increase of ash content; that material is made from polyethylene, paper, and also aluminum, which likely increases an ash content even more. On the other hand, for materials such as fabrics, kitchen waste, and wood, an increase of CSF’s HHV is promoted with an increase in process temperature (Figure 3).

These differences in results indicate that there is no one optimal pyrolysis condition for RDF pyrolysis. Therefore, the determination of optimal conditions of process temperature and time should be determined based on the morphological analysis of RDF by taking into account the optimal conditions for waste with the greatest fraction. It means that the third variable (the RDF component share) should also be optimized.

### 3.2. General Model of HHV of Carbonized RDF

The general model of HHV of carbonized RDF mixtures was used to estimate the HHV of two carbonized RDF blends (RDF 1 and RDF 2). In this way, the general model for the RDF mixture was validated by the application of the experimental data. The results of correlations between experiment and model results were presented in Figure 4. The results showed that the proposed model (Equation (3)) explains 75% and 70% of the data variability (determination coefficients R^2^ = 0.75 and 0.70) for RDF blend 1 and blend 2, respectively (Figure 4). The root means square errors were 5.03 and 6.30 for RDF blend 1 and RDF blend 2, respectively (Appendix B, Table A1). The obtained results show that it is possible to predict the HHV of CSF produced from RDF on the basis of the assumptions of the pyrolysis temperature, residence time, and the knowledge of the RDF composition share. This finding has the following practical implications:It is possible to fit the pyrolysis parameters to the local RDF properties/share of main components to obtain the highest HHV;The RDF composition should also be optimized by the mechanical sorting of raw MSW to obtain the CSF with high-quality fuel properties.

As this model was determined based on the eight common RDF components, it could be further improved with the addition of more types of RDF components, ash content, and also with the application of the probabilistic approach (artificial networks) instead of the deterministic approach.

## 4. Conclusions

Executed experiments showed that in the case of paper, cardboard, and kitchen waste, there was no significant influence of the pyrolysis temperature and process time on the resulting HHV of CSF. In the case of wood and fabrics, the increase in process temperature increased the HHV of the CSF. Plastics and rubber reacted oppositely, leading to a decrease of CSF HHV with the increase of pyrolysis temperature. In the case of composite PAP/AL/PE material, the temperature of 420 °C with the highest HHV of CSF 30 MJ·kg^−1^ was found. The findings for particular constituents of RDF and composite material PAP/AL/PE show that depending on the type of the material and its composition, the final calorific value of the RDF pyrolysis solid product may differ, depending on the RDF composition.

It is recommended that the estimation of the CSF calorific value for the specific waste processing site is completed by modeling rather than by the flat rate assumption from other locations. This could aid in improving the decision-making for technology upgrades. Therefore, we proposed the technical model, which may be useful in the quick prediction of the HHC of the CSF depending on the RDF composition of its main components and technical parameters of the pyrolysis.

The model enables finding an optimal condition of the pyrolysis process for CSF production, depending on the available local waste mixture. The experimental model validation showed that it explains 75% and 70% of the data variability when applied to RDF mixtures. As this approach has been proposed for the first time, it requires further improvement by generating data on the additional waste material pyrolysis, other RDF types, and by application of other probabilistic prediction methods, such as neural networks or fuzzy logic models.

Executed experiments revealed practical implications for conducting the pyrolysis process depending on the RDF composition when the production of high-quality CSF is the main goal. The RDF should be processed at as low as possible pyrolysis temperature (300 °C) to obtain the CSF with the highest HHV when paper, carton, plastic, and rubber dominate the RDF mixture. In comparison, when materials such as fabrics and wood dominate the RDF mixture, they should be pyrolyzed at a higher temperature (500 °C) to obtain greater HHV.

The proposed model may be applied for the optimization of the existing pyrolysis plants treating the RDF. The best condition for CSF production is highly dependent on the most abundant waste in the waste mixture; therefore, before CSF production, a simple waste morphological analysis is recommended. By using proposed models, it is possible to fit the pyrolysis parameters to the site-specific RDF properties to obtain the CSF with the desired high-quality fuel properties. In the case of the existing pyrolysis plant with limited options for the process modification, the models allow finding the optimal RDF composition mix that could be implemented via the mechanical sorting treatment of MSW before the pyrolysis.

## Figures and Tables

**Figure 1 materials-14-00049-f001:**
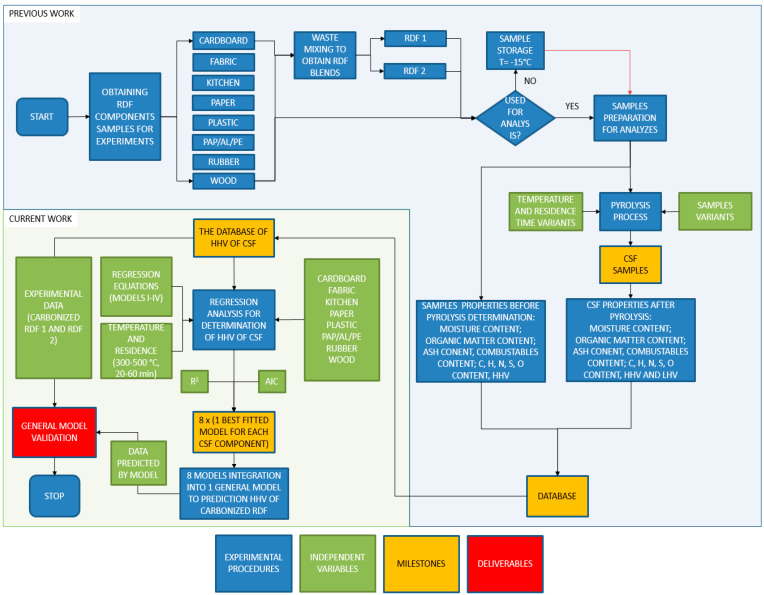
The experimental procedure flowchart for modeling and optimizing the higher heating value of solid carbonized fuel generated from blends of refuse-derived fuel by low-temperature pyrolysis.

**Figure 2 materials-14-00049-f002:**
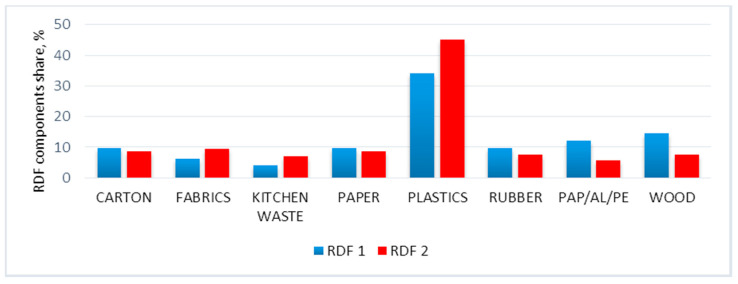
Percentage of individual major waste groups in refuse-derived fuel (RDF) 1 and RDF 2 [58].

**Figure 3 materials-14-00049-f003:**
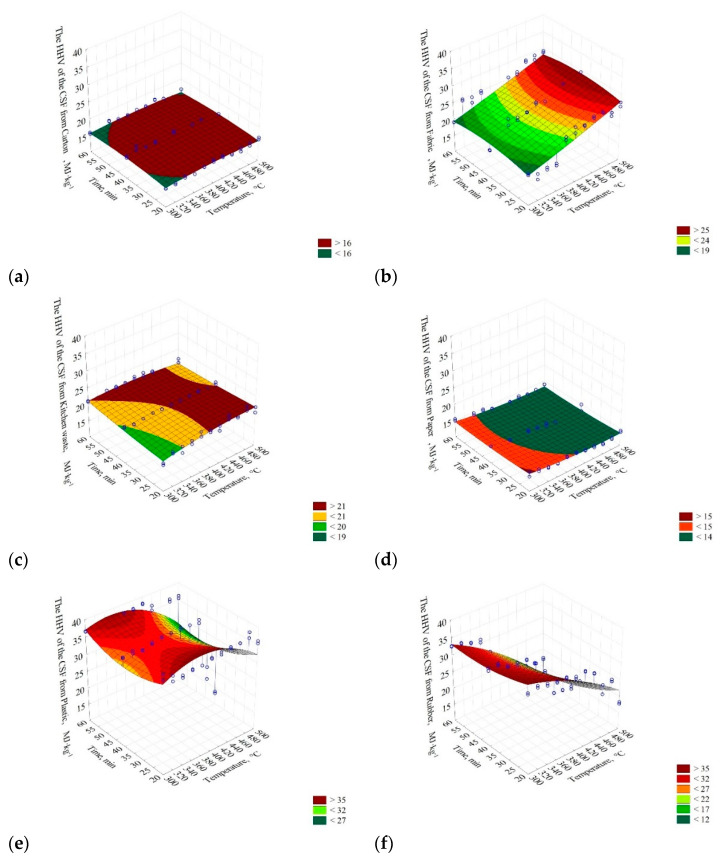
The influence of low-temperature pyrolysis temperature and residence time on the high heating value of CSF from (**a**) carton; *HHV(T,t)* = 6.99959 + 0.0376341·*T* − 4.51616·10^−5^·*T*^2^ + 0.0958599·*t* − 0.00121729·*t*^2^, (**b**) fabric; *HHV(T,t)* = 5.27185 + 0.0254657·*T* + 2.32626∙10^−5^·*T*^2^ + 0.256539·*t* − 0.00309588·*t*^2^, (**c**) kitchen waste; *HHV(T,t)* = 1.95489 + 0.0758155·T + 0.127447·t − 6.77225·10^−5^·T^2^ + 0.00050108·t^2^ − 0.000392949·T·t, (**d**) paper; *HHV(T,t)* = 26.6224 − 0.0489416·T + 5.21567·10^−5^·T^2^ − 0.0871204·t + 0.000907272·t^2^, (**e**) plastic; *HHV(T,t)* = −26.9986 + 0.383563·T − 0.358936·t − 0.000485497·T^2^ + 0.00735668·t^2^ − 0.000682253·T·t, (**f**) rubber; *HHV(T,t)* = 28.781 + 0.100862·T − 0.0882796·t − 0.000193804·T^2^ + 0.00338057·t^2^ − 0.000857182·T·t, (**g**) PAP/AL/PE composite packaging pack; *HHV(T,t)* = −124.359 + 0.711538·T + 0.648268·t − 0.00079769·T^2^ + 0.000440254·t^2^ − 0.00174132·T·t, (**h**) wood; *HHV(T,t)* = 1.36811 + 0.0757645·*T* + 0.157354·y − 5.58308∙10^−5^·*T*^2^ − 0.000491547·t^2^ − 0.000220373·*T*·*t*. The chosen best-fitting mathematical models of the influence of the pyrolysis temperature and residence time on the HHV of CSF produced from different RDF components are given.

**Figure 4 materials-14-00049-f004:**
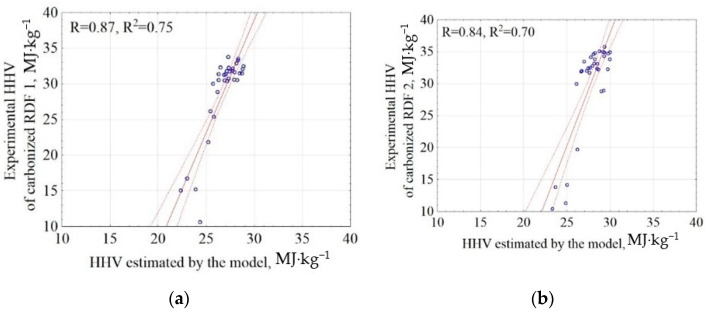
Correlation between experimental and estimated HHV of carbonized RDF blends, (**a**) CSF from RDF blend 1; RDF 1, (**b**) CSF from RDF blend 2; RDF 2. Blue circles are points common for experiments and models.

**Table 1 materials-14-00049-t001:** Advantages and disadvantages of methods for predicting the higher calorific value [39,40,41,52,54,55].

HHV Prediction Methods	Advantages	Disadvantages
Equations based on proximate analysis	- the simplest models, easy to use - data easy to get—the simplest equipment and cheapest analytical methods	- there is a need to make corrections for different materials - there are a lot of different standards for proximate analysis (eq. EN/ISO/ASTM/TGA) that can be used to perform analysis (differences may lead to wrong predictions if the model was not correct) - not precise as other methods
Equations based on elemental analysis (C, H, N, S, O)	- the simplest models, easy to use - the possibility of obtaining fairly accurate results	- the data are much more expensive to get—laboratory analysis require advanced equipment (elementary analyzer)
Neural networks	- solving problems without knowing the analytical dependency between inputs and expected outputs - generalization ability	- laboratory analysis—time-consuming because large amounts of data are needed, - special software needed for data analysis - the danger of overtraining or insufficient training of the network - limited possibilities to make conclusions about relations

**Table 2 materials-14-00049-t002:** Regression models applied to describe the dependence of higher heating value (HHV) of carbonized solid fuel (CSF) produced from different RDF components to the pyrolysis temperature (*T*) and residence time (*t*).

Regression	Model Equation
(I) linear model	*HHV = a_1_ + a_2_·T + a_3_·t*
(II) second-order polynomial model	*HHV = a_1_ + a_2_·T + a_3_·T^2^ + a_4_·t + a_5_·t^2^*
(III) factorial regression	*HHV = a_1_ + a_2_·T + a_3_·t + a_4_·T·t*
(IV) quadratic regression	*HHV = a_1_ + a_2_·T + a_3_·t + a_4_·T^2^ + a_5_·t^2^ + a_6_·T·t*

where: *a*_1_—intercept; *a*_2_*–a*_6_—regression coefficient; *T*—temperature, *T* = 300–500 °C; *t*—time, *t* = 0–60 min.

**Table 3 materials-14-00049-t003:** Results of regression models of HHV of CSF determined from RDF components in relation to pyrolysis temperature and residence time.

Material	Assessment Criterion	Model			
		I	II	III	IV
Carton	*R^2^*	0.02	*0.14*	0.02	0.14
	*AIC*	250.89	*245.98*	252.70	247.77
Fabric	*R^2^*	0.51	*0.53*	0.52	0.53
	*AIC*	415.75	*416.63*	416.68	417.51
Kitchen waste	*R^2^*	0.18	0.24	0.35	*0.41*
	*AIC*	267.39	265.66	254.07	*250.75*
Paper	*R^2^*	0.29	*0.36*	0.29	0.36
	*AIC*	252.93	*250.05*	254.82	251.93
Plastic	*R^2^*	0.16	0.32	0.18	*0.34*
	*AIC*	493.92	483.38	494.58	*483.70*
Rubber	*R^2^*	0.84	0.86	0.85	*0.87*
	*AIC*	415.03	410.78	409.69	*404.39*
PAP/AL/PE composite packaging pack	*R^2^*	0.01	0.51	0.21	*0.72*
	*AIC*	463.92	421.00	450.61	*386.88*
Wood	*R^2^*	0.76	0.78	0.78	*0.78*
	*AIC*	258.47	257.80	255.36	*254.29*

*Italic* font signifies that model chosen to further analysis. Higher *R^2^* values and lower AIC = better model.

## Data Availability

The data presented in this study are available in Appendix A.

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
