# Peer review of "The Prediction of Calorific Value of Carbonized Solid Fuel Produced from Refuse-Derived Fuel in the Low-Temperature Pyrolysis in CO2"

_materials, 2020, doi:10.3390/ma14010049_

Round 1
Reviewer 1 Report
This paper presents numerical modelling results about Calorific Value of CSF Produced From RDF in the Low-Temperature Pyrolysis in CO2. This finding can have implications in the energy industry. This is a well-written communication and I would recommend it to be published after some minor modifications as noted below.
- Line 40: it is worth to add some other examples of residual waste upgrade methods or refer to a review article.
- The connection between some paragraphs are missing in the introduction, I would recommend revising the introduction to have a clear story.
- Figure 1: please use a bigger font size as some part of the diagram is not clear.
- Experimental: if all the results have been published previously and the experimental procedure explained in the previous article. I think you can remove the experimental section to avoid confusion and duplication.
- 182-185: I would suggest putting the number in a table or a graph so it will be easier to compare.
- Equation 4: I am not sure if it is necessary to keep this and similar equations in the manuscript as they are simple mathematic equations. However, I leave the decision to the authors.
- I think the results and discussion is been written properly. However, the authors need to add more details about the implications of their finding. I wouldn’t recommend publishing the article without addressing this comment as it would be only fitting several simple equations to previously published experimental data.
Reviewer 2 Report
This work investigates the prediction of calorific value of carbonized solid fuel derived from waste material as pyrolysis feedstock. Four mathematical functions linear, second-order polynomial, factorial regression, and quadratic regression were developed and evaluated for their potential in accurate estimation of heating value for different blends. Obviously, the best condition for CSF production was highly dependent on the most abundant waste in the mixture. Even though the concept of this research offers fundamental interest to the field, the present version of this manuscript is not suitable for publication in Materials and require major revision. Here are my points of criticism that must be addressed before reconsideration for publication:
1- In the abstract, what do PAP, AL, and PE stand for? Even in the introduction (line 88) this information is not provided. Abbreviations must be expanded as they appear first in the manuscript. Besides, some key findings of the research must be mentioned clearly in the abstract.
2- Abstract: “… best model among four mathematical functions …. Next, these 8 models were integrated …”. Please make necessary correction. Is this work studying four models or eight models?
3- Accurate prediction of thermal conversion product properties (such as product distribution, chemical composition, energy content, etc.) and process variables is a huge challenge in the field of renewable energies. Thus, the application of computing, modelling, and simulation tools has become a point of interest for research community. The current Introduction includes large amount of basic background information and does not highlight most promising techniques for mathematical modelling of thermochemical processes. The Introduction must be improved by discussing other related publications to provide broad information on recent progress. Here are some most up-to-date papers that must be mentioned in the revised manuscript:
- Biomass gasification in a downdraft fixed-bed gasifier: Optimization of operating conditions. Chemical Engineering Science, 2020, https://doi.org/10.1016/j.ces.2020.116249.
- Experimental and modelling study of the torrefaction of empty fruit bunches as a potential fuel for palm oil mill boilers. Biomass and Bioenergy, 2020, https://doi.org/10.1016/j.biombioe.2020.105530.
- Application of Artificial Intelligence in the Prediction of Thermal Properties of Biomass. Valorization of Biomass to Value-Added Commodities, Green Energy and Technology, 2020, https://doi.org/10.1007/978-3-030-38032-8_4.
The revision must clearly discuss advantages/disadvantages of the present methodology compared to each article mentioned above.
4- Figure 1 is not explained properly. I would recommend that the authors clearly discuss the experimental matrix and be more verbose on data collection and processing. Additionally, Akaike analysis must be described in details.
5- Prediction of heating value based on elemental analysis (CHNS-O elements) have been well-stablished in the past. How would such models (such as Dulong’s equation, Boie equation, etc.) differentiate HHV prediction from mathematical models presented in this work. For more details I would refer the authors to “Introduction to Biomass Energy Conversion” by Sergio C. Capareda, CRC Press, Taylor and Francis Group.
6- Authors have concluded that the HHV of the carton did not change significantly with the increase temperature and pyrolysis process time and was around 16 MJ∙kg-1. Assuming performing two experiments at higher and lower ends (30 min, 330°C versus 60 min, 500°C) this conclusion hardly makes practical sense. Therefore, the limitations of each model should be mentioned clearly.
7- In the case of the PAP/AL/PE composite packaging pack (Figure 2g) the HHV behavior have been blamed on the lack of secondary reactions. What kind of secondary reactions can contribute to this strange trend?
8- Apparently, ach RDF component and RDF blends were characterized for moisture content, organic matter content, loss on ignition, ash, combustible content, elemental compositions (CHNS-O), and HHV. Although a previous publication have been cited for these data, I would recommend that the authors present this information in the revised manuscript, or in the Appendix (if there are limitations for the number of Tables in this communication paper).
9- Conclusion must be re-written and provide key findings of the research rather than being too general.
Round 2
Reviewer 2 Report
Accept for publication.
This manuscript is a resubmission of an earlier submission. The following is a list of the peer review reports and author responses from that submission.
Round 1
Reviewer 1 Report
The work is not a scientific breakthrough, but could be useful since it aims at providing an easy tool for the prediction of HHV of RDF as a function of pyrolysis conditions, however it cannot be accepted in its presnet form becasue it is written poorly. English language must be carefully checked. I tried to make some corrections, but I think that a professional editing would be appropriate.
The introduction is too long with too many data.
In the attached file you find additional comments:
Why does HHV of some components decrease above certain temperature?
Did you check if combustion had occurred by any chance?

Author Response
Thank you very much for your time and feedback. We addressed all issues itemized below.
|
Reviewer 1 |
||
|
# |
The Comments and Suggestions |
The author's answers |
|
1 |
The work is not a scientific breakthrough, but could be useful since it aims at providing an easy tool for the prediction of HHV of RDF as a function of pyrolysis conditions, however it cannot be accepted in its presnet form becasue it is written poorly. English language must be carefully checked. I tried to make some corrections, but I think that a professional editing would be appropriate. |
We appreciate these comments. We addressed them and significantly improved the manuscript. |
|
2 |
The introduction is too long with too many data. |
Thank you for that comment. We removed a part of the information describing a particular waste type. However, another reviewer asked for providing more information in the Introduction. We tried to strike a balance here. We made a significant effort to cover all the necessary data to show the importance of this study, and we tried to focus on the most relevant aspects related to the manuscript topic. |
|
3 |
In the attached file you find additional comments: |
We are thankful for pointing out grammar and other issues. We addressed comments. Regarding comments from the file, we used the following format: · italics - text in the original version of the manuscript, · green – reviewer comments, · red – author's response. - Line 163: “The goal was to provide eight models that quantitatively describe the impact of temperature and residence time of pyrolysis on the high heating value of CSF produced from different components of RDF.” >>> why 8? >>> The misleading number of models were removed. There were 8 models, 1 model for each of the RDF components.
- Line 168 >>> Typos in Figure 1 were corrected;
- Line 236-238 “The data calculated by models were validated using a linear correlation with the experimental data of two carbonized RDF blend, with the significance level p <0.05, and by the root mean square error (RMSE). The RMSE is the standard deviation of the residuals and is given by equation 3:” >>> something missing in this sentence, moreover in previous parts you used R2. What was R2 then? And p? You do not define it anywhere. >>> The sentence was replaced as follows “The models were validated by using a linear correlation with the experimental data of two carbonized RDFs. The values predicted by the model were compared with the experimental data using a root mean square error (RMSE). The RMSE is the standard deviation of the residuals, Equation 4:” The definition of R2 was added in the lines 206-215. We did not check the significance for model coefficients. The information about that was removed – it was an artifact rewritten from our other methodology by mistake. Here, the p-value was used to state if the significant changes in particular models were observed, i.e., to verify if temperature and time have an impact on HHV. When the p-value calculated by the statistical analyses software was lower than 0.05, we assumed that differences were significant (i.e., the temperature or time had an impact on the HHV of CSF).
- Line 298 (Figure 2) >>> Turn the description of the z-axis upsidedown >>> The direction of the description of the z-axis is generated by the software automatically, and we cannot change it.
- Line 335 (Figure 3) >>> what are these hystogram? >>> The histograms show how many measurements were observed for a particular interval of HHV. The vertical (y-axis) histograms present a number of observations for particular HHV measured in the experiment, and on the horizontal (x-axis) for HHV estimated by the model. |
|
4 |
Why does the HHV of some components decrease above a certain temperature? |
Line 307-308 “temperature leads to HHV decreasing” >>> We added a possible explanation of that phenomena as follows (lines 322-343): “The decrease in the case of plastic (polyethylene) material, and likely rubber as well, could be a result of pyrolysis performed in an open reactor (the pyrolysis gases were allowed to vent from the rector), and therefore, secondary reactions that favor biochar production did not take place. Tiikma et al. [44] reported that during the pyrolysis of polyethylene in a closed reactor, biochar yield decreased from 420 °C to 440 °C, while > 440 °C it started to increase, whereas a liquid fraction showed the opposite trend. It means that the liquid starts to decompose to biochar and gas (secondary reactions) above 440 °C. It is worth noting that these reactions took place after ~90 min, and continued with longer residence time [44]. Knowing that CSF produced for this study had a long cooling period from setpoint temperature to ambient (several hours), we assume that the reason for HHV decrease was due to the lack of secondary reactions. The decrease of paper and carton (cellulose-based materials) HHV at higher temperatures is also interesting. Several works on the pyrolysis of cardboard and paper were investigated [45–47]. Results show that HHV of CSF from cardboard/paper slightly increases with temperature increase, but above a certain temperature, it starts to decline [45,46]. The reason for that could be due to the high ash content that increases with the process temperature [46]. The residence time also has a significant impact on HHV, in the above-mentioned works, materials were pyrolyzed at residence time longer than 30 min. In work of Sotoudehnia et al. [47], a short residence time was used (6.4 s), and the increase of pyrolysis temperature resulted in a slight increase of HHV for the CSF from cardboard. Additional reason of that phenomenon could be the loss of carbon due to formation of acetic acid and its volatilization. In the case of the PAP/AL/PE composite packaging pack (Figure 2g) the HHV behavior may be related to lack of secondary reactions and to increase of ash content; that material is made from polyethylene, paper, and also aluminum, which likely increases an ash content even more” |
|
5 |
Did you check if combustion had occurred by any chance? |
All materials were pyrolyzed at the same time in the separate crucibles at a particular temperature and residence time without access of air. For this reason, this is unlikely. |
Reviewer 2 Report
The paper is of scientific and original nature, related to The Prediction of Calorific Value of Carbonized Solid Fuel Produced from Refuse-Derived Fuel in the Low-Temperature Pyrolysis.
For a better clarification, please edit your paper as follows:
Enlarge the Introduction with current results reported in the world and Europe, - References to expand the results of European authors registered in SCOPUS / WoS such as: Pyrolysis conversion of polymer wastes to noble fuels in conditions of the slovak republic. The title of paragraph begins at the bottom of the page (page 3 – line 146). Figure 1 - should be contrasting and readable (change green color), conclusions and future work should be extended to contain practical applications based on research described in this paper, edit the paper according to the template, modify the mathematical expression (formula) No: 2. Table A1 is inappropriately divided on two pages, then difficult to read for readers.
Please, edit the paper according to previous comments and after minor changes I recommend the paper to be published.
Author Response
Response to Reviewer 2 Comments
Thank you very much for your time and feedback. We addressed all issues itemized below.
|
Reviewer 2 |
||
|
# |
The Comments and Suggestions |
The author's answers |
|
1 |
The paper is of scientific and original nature, related to The Prediction of Calorific Value of Carbonized Solid Fuel Produced from Refuse-Derived Fuel in the Low-Temperature Pyrolysis. For a better clarification, please edit your paper as follows: |
We are thankful for these comments. We addressed them below. |
|
2 |
Enlarge the Introduction with current results reported in the world and Europe, - References to expand the results of European authors registered in SCOPUS / WoS such as: Pyrolysis conversion of polymer wastes to noble fuels in conditions of the slovak republic. |
We enlarged the Introduction with current results from Europe. Please see lines 70-82. We had to balance our approach to revising the Introduction since Reviewer 1 requested a reduction of the information presented in this section. |
|
3 |
The title of paragraph begins at the bottom of the page (page 3 – line 146). |
The title of the paragraphs was moved to the start of page 3 |
|
4 |
Figure 1 - should be contrasting and readable (change green color); |
Figure 1 was improved. We change the white font to black.
|
|
5 |
Conclusions and future work should be extended to contain practical applications based on research described in this paper; |
The Conclusion was enlarged with practical applications based on research results as follows: Line 380-384 “By using proposed models, it is possible to fit the pyrolysis parameters to the site-specific RDF properties to obtain the CSF with high-quality fuel properties. In the case of the existing pyrolysis plant with limited options for the process modification, the models allow finding the optimal RDF composition mix that could be implemented at the step of the mechanical sorting of MSW.” |
|
6 |
Edit the paper according to the template; |
The paper was edited according to the template |
|
7 |
Modify the mathematical expression (formula) No: 2; |
We do not see a problem with formula No. 2 |
|
8 |
Table A1 is inappropriately divided on two pages, then difficult to read for readers; |
Table 2 was moved to be showing on one page. |
|
9 |
Please, edit the paper according to previous comments and after minor changes I recommend the paper to be published. |
Thanks. We addressed comments. |
Round 2
Reviewer 1 Report
I appreciated the revision done to the text by the authors, but I must reject the paper due to a serious technical error.
I had noticed, already in my first revision, that the HHV of the char decreased for high temperature and times, especially for plastics and I had asked to explain it. I had also asked if combustion could have happened.
The authors replied that no air leakcage was possible. However they said that the ash content increased ("Results show that HHV of CSF from cardboard/paper slightly increases with temperature increase, but above a certain temperature, it starts to decline [45,46]. The reason for that could be due to the high ash content that increases with the process temperature [46]...".).
This made me suspicious, again, that you have consumed fixed carbon. So I went to read the full experimental procedure from ref 43, and downloaded the supplemetary files of that reference. Here I learnt that: 1. you are not in inert atmosphere: you have CO2 atmospheres, which makes it possible to have gasification! 2. The ash content inreases up to 50% even for plastics. This really tells me that you have not pyrolysed, but rather pyro-gasified the samples!!!!